# Early Steps of Hepatitis B Life Cycle: From Capsid Nuclear Import to cccDNA Formation

**DOI:** 10.3390/v13050757

**Published:** 2021-04-26

**Authors:** João Diogo Dias, Nazim Sarica, Christine Neuveut

**Affiliations:** Laboratoire de Virologie Moléculaire, Institut de Génétique Humaine, CNRS, Université de Montpellier, UMR9002 Montpellier, France; joao.dias@igh.cnrs.fr (J.D.D.); nazim.sarica@igh.cnrs.fr (N.S.)

**Keywords:** HBV, HBVcccDNA, nuclear import, nuclear pore, DNA repair, DNA synthesis, HBV cure, HBc

## Abstract

Hepatitis B virus (HBV) remains a major public health concern, with more than 250 million chronically infected people who are at high risk of developing liver diseases, including cirrhosis and hepatocellular carcinoma. Although antiviral treatments efficiently control virus replication and improve liver function, they cannot cure HBV infection. Viral persistence is due to the maintenance of the viral circular episomal DNA, called covalently closed circular DNA (cccDNA), in the nuclei of infected cells. cccDNA not only resists antiviral therapies, but also escapes innate antiviral surveillance. This viral DNA intermediate plays a central role in HBV replication, as cccDNA is the template for the transcription of all viral RNAs, including pregenomic RNA (pgRNA), which in turn feeds the formation of cccDNA through a step of reverse transcription. The establishment and/or expression of cccDNA is thus a prime target for the eradication of HBV. In this review, we provide an update on the current knowledge on the initial steps of HBV infection, from the nuclear import of the nucleocapsid to the formation of the cccDNA.

## 1. Introduction

The human hepatitis B virus (HBV) remains a major health threat with an estimated 250 million people chronically infected worldwide [1]. Epidemiological studies have established that persistent HBV infection is a major risk factor for the development of hepatocellular carcinoma (HCC), making HBV one of the most important environmental carcinogens for humans [2,3]. HBV is the prototype member of a family of small, enveloped DNA viruses called hepadnaviruses that preferentially infect hepatocytes. All hepadnaviruses have the ability to replicate their genomes in the cytoplasm through reverse transcription of the encapsidated pregenomic RNA (pgRNA) by the viral reverse transcriptase (Pol; Figure 1). This enzyme has DNA-dependent DNA polymerase and RNA-dependent DNA polymerase activities and RNase H activity that generate a partially double-stranded DNA genome called relaxed circular DNA (rcDNA) [4].

The 3.2 kb rcDNA is enclosed in an icosahedral nucleocapsid that is enveloped to form infectious particles. Upon infection, the rcDNA intermediate form is repaired so as to give rise to a complete double stranded covalently closed circular DNA (cccDNA) that is located in the nucleus and serves as a template for the transcription of all viral RNAs, including pgRNA [5,6]. In the cytoplasm, after reverse transcription, mature capsids containing rcDNA assemble with the viral envelope at the cellular multivesicular body, allowing for the formation and secretion of viral particles (Figure 1) [7,8,9]. Alternatively, capsids can return directly to the nucleus, resulting in intracellular cccDNA amplification [10,11]. The mechanism that directs the viral capsids towards either of these two pathways is still unknown. Intracellular recycling appears early after infection, before virus secretion, for the related virus duck hepatitis B virus [12]. It has thus been suggested that the fate of the capsids is controlled by the level of viral envelope proteins [13]. The establishment of cccDNA in the nucleus requires the completion of different steps, starting with the attachment of the viral particle to the cell surface, followed by the delivery of the nucleocapsid into the cytoplasm, the translocation of the rcDNA into the nucleus, and its repair to form the cccDNA. The exact chronology of the two last steps is not yet fully understood and may overlap, as will be discussed below. Of note, besides the secretion of viral particles that consist of rcDNA-containing virions, different types of incomplete particles are released, particularly empty capsids consisting of genome-free enveloped capsids that are found in infected humans in 100-fold or more excess [14]. These empty viral particles are currently the subject of research by different groups, as they represent a potential biomarker to monitor intrahepatic HBV activities and the response to antiviral therapies [15,16,17]. HBV RNA-containing capsids are also secreted as both non-enveloped and enveloped particles [18,19].

Current treatments for chronic hepatitis B efficiently control virus replication and improve liver function, but do not lead to complete viral clearance, and lifelong treatment is required for these patients [20]. Therapeutic failure is due, in part, to the inability of the drugs to eliminate the cccDNA, making this the key target in order to achieve HBV cure. Understanding HBV cccDNA biology from the entry of viral particles to the establishment of cccDNA, as well as its transcriptional regulation and persistence will allow for the identification of new therapeutic targets to clear the viral reservoir, thus representing a very active field of investigation. In this review, we summarize and provide an update on early events of HBV infection, with a focus on the critical steps from the nuclear import of the nucleocapsid to cccDNA formation.

## 2. Virus Entry

The viral envelope contains three viral surface proteins: the large hepatitis B virus surface protein L, the middle hepatitis B viral protein M, and the small hepatitis B virus surface protein called S. HBV first attaches with a low affinity to hepatocytes via binding to heparan sulfate proteoglycans (HSPGs), which is subsequently followed by the binding of L to the high affinity receptor sodium taurocholate co-transporting polypeptide (NTCP). The study of HBV internalization and the mechanisms regulating it were severely limited for many decades, as the identification of NTCP as the HBV receptor was only published nine years ago [21]. While HBV enters the cell via endocytosis, the underlying mechanism, whether it is caveola- or clathrin-dependent, is still debated [22,23]. Recent studies support the latter as the major mechanism involved in HBV entry [24,25]. Moreover, it has been shown that the machinery involved in the endocytosis of epidermal growth factor receptor (EGFR), which acts as a cofactor for HBV entry, is also involved in the internalization and transport in the endosomal network of HBV [26,27]. The mechanisms leading to HBV escape from endosomes are not yet fully understood, and membrane fusion may be mediated by the fusogenic domains identified in the different surface proteins L, M, and S [7,28].

## 3. Intracellular Transport and Nuclear Import

Once released in the cytoplasm, the HBV nucleocapsid will traffic toward the nucleus in order to release the viral genome. The HBV capsid is composed predominantly of 120 dimers of the HBV core protein (HBc), which can assemble into a T = 4 icosahedral symmetry [9]. HBc is 183 amino acids long (or 185 depending of the virus genotype) and consists of two separate domains—the N-terminal domain (NTD) responsible for the formation of the capsid shell and the C-terminal domain (CTD), rich in arginine residues and essential for interaction with nucleic acids. CTD is involved in pgRNA encapsidation and DNA maturation during HBV replication [29,30,31]. However, this functional separation of HBc is over simplified, as different studies have shown that NTD plays a role in DNA synthesis and, conversely, CTD participates in capsid assembly [32,33]. CTD is highly basic and undergoes dynamic phosphorylation and dephosphorylation, which regulate many HBc functions including pgRNA encapsidation, reverse transcription, capsid stability, and intracellular trafficking [34,35,36,37].

As HBV infects non-proliferating hepatocytes, the entry of the HBV genome into the nucleus requires transit through the nuclear pores complex (NPC), which serves as a gatekeeper to the nucleus. NPC is composed of multiple copies of a large number of proteins called nucleoporins (Nups), forming a large structure that spans the double-membraned nuclear envelope. NPC forms a channel that allows small proteins (less than 40 kDa) to diffuse passively, while excluding larger ones. The nuclear import of these larger proteins involves their interaction with nuclear import receptor members of the importin (Imp)/karyopherin family. This interaction is induced by the recognition of a nuclear targeting signal on the surface of the cargo protein by the import receptor [38].

Like other viruses, HBV exploits the microtubular network for efficient nuclear delivery in order to overcome the high viscosity of the cytoplasm. Moreover, microtubule-dependent movement may provide a direct and efficient route to the NPC and the pool of importin proteins localized at the nuclear periphery [39]. Studies have shown that the destabilization of microtubules with nocodazole blocks the import of the HBV capsid, as well as the related virus duck hepatitis B virus (DHBV) [40,41]. Recently, the group of M. Kann further confirmed the requirement of a microtubule for HBV nuclear trafficking, and showed that the HBV capsid uses the dynein motor complex for nuclear trafficking via an interaction with the dynein light chain LL1 [42].

The mechanism leading to rcDNA nuclear import remains a matter of debate, and two different models have been proposed. In the first model, HBV capsids are believed to be responsible for NPC docking and genome delivery. Indeed, the CTD region, which is highly conserved between all genotypes, contains nuclear localization signal (NLS) sequences that have been shown to be functional when using NLS defective SV40 large T antigen as a reporter for subcellular localization or by studying the impact of mutations on HBc subcellular localization [35,43,44]. The role of these sequences in nuclear trafficking was further confirmed in the context of an assembled capsid by performing competition experiments for NPCs binding with peptides corresponding to NLS, using digitonin-permeabilized cells, or by studying capsid subcellular localization in hepatoma cells cotransfected with a vector coding HBc containing mutations in the putative NLS and viral polymerase [43,44,45]. Further studies using coimmunoprecipitation assays or nuclear binding assays and competition assays with wheat germ agglutinin (WGA), which blocks active nuclear import by nuclear transport receptors using permeabilized cells, showed that the docking of capsids to the nuclear pore is mediated through their interaction with transport receptors importin α (Imp α) and importin β (Imp β) [45,46,47]. Of note, the exact position of the NLS sequences is still debated, but seems to overlap with two of the arginine rich domains at residues 150–152 and 165–168 [43,44,45]. These observations imply a conformation in which the HBc CTD containing the NLS should be exposed at the exterior side of the capsid. The exposition of CTD may vary depending on the status of HBc phosphorylation, the nature of the nucleic acid present, and the step of genome maturation [36,45]. Indeed, biochemical and cryo-electron microscopy structural studies have shown that, depending on the presence or not of RNA (empty or pgRNA filled capsids) and the state of phosphorylation (using phosphomimetic CTD mutants), CTD is differentially exposed [32,44,45]. CTD is thus transiently exposed at the exterior in an empty capsid, while the negatively charged RNA retains the basic CTDs inside the capsid [48,49]. However, the phosphorylation of the RNA-filled capsid has been shown to induce nuclear pore binding, supporting the hypothesis that phosphorylation modulates CTD exposure [45]. The capsid of mature HBV particles (rcDNA filled particles) appears to be mainly dephosphorylated, which could theoretically lead to the retention of the CTD in the interior. However, the maturation process, by which RNA is retrotranscribed into dsDNA, is associated with structural changes. Mature capsids are thus more unstable, in part due to the nature of DNA, which is less flexible and much less able to interact with CTDs [50,51]. CTD could thus be exposed at the exterior, which may be facilitated during capsid breathing [52,53]. Moreover, as already mentioned, phosphorylation is a dynamic process that can also be involved in CTD exposure in mature capsids. Recently, Luo and colleagues described the phosphorylation of a mature capsid that occurs upon virus entry or capsid recycling, and that could modulate trafficking, uncoating, and cccDNA formation [54]. Finally, CTD exposure in mature particles is supported by studies showing that mature capsids are uniformly cleaved by trypsin, contrary to immature particles, which remain partially protected from digestion [47]. While pgRNA containing capsids have been shown to be largely excluded from NPCs, likely because CTDs are on the capsid interior and are thus not available for importin interaction, cryo-EM studies have shown that CTDs in empty capsids are exposed at the exterior of the capsids [47,49]. However, in empty capsids, CTDs interact directly with importin β via an importin β-binding sequence (IBB) located between amino acids 141–180, suggesting a distinct intracellular trafficking from that of mature capsids [55]. Whether importin β is responsible for the trafficking of empty capsids and capsid destabilization, as suggested in this study, requires further investigation.

After their docking to NPCs, HBV capsids proceed to the nuclear basket where uncoating occurs via capsid disassembly. HBV DNA is then released into the nucleoplasm and HBc dimers, which can reassociate to form capsids [56]. In the course of their study, Kann and collaborators observed an unusual feature of nuclear import: although both immature and mature capsids interact with the NPC and diffuse in the nuclear basket, only mature capsids disintegrate while immature capsids remain trapped. They showed that this retention is mediated by direct interaction of the viral capsid NTD with a protein of the nuclear basket: nucleoporin 153 (Nup 153). Interestingly, mature capsids, while binding to Nup153, disintegrate, allowing capsid dimers and viral DNA to reach the nucleoplasm (Figure 2) [57]. This direct binding of the capsid to Nup153 could explain the fact that Ran GTP is not required for the release of viral DNA, even if the capsid was bound to importin α/β during the first steps of translocation.

What is the benefit of Nup153 binding if capsids enter via importin α/β transport? Is transient binding of capsids to Nup153 required for their disintegration? Indeed, disassembly seems to occur in the nuclear basket as UV cross linking of capsids does not impair the localization of capsids in the basket, while it impairs the HBV DNA nuclear translocation [57]. The mechanisms leading to uncoating are not fully understood. As mentioned earlier, mature capsids are less stable than immature capsids [50]. In vitro experiments have shown that dsDNA-filled capsids are less stable than capsids associated with RNA [9,51]. DNA synthesis may contribute to capsid destabilization because of the nature of DNA itself, which is more rigid. Post-translational modification of the capsid, such as phosphorylation, may also facilitate uncoating [30,32,33,50]. Indeed, capsid stability is believed to be maintained through the establishment of balanced electrostatic interactions in the capsid, which can be modulated by phosphorylation/dephosphorylation events [37]. Additionally, host cellular factors may also be involved in the selective process of mature capsid uncoating. Indeed, mouse cells do not support HBV cccDNA establishment. However, one mouse hepatoma cell line has recently been shown to accumulate HBV cccDNA, which correlates with the appearance of unstable capsids in the cytoplasm [58]. Considering that the inability of mouse cells to support cccDNA formation is not due to the expression of a murine restriction factor, but more likely to the absence of cellular factors, we can hypothesize that HBV uncoating also requires the activity of host factors that remain to be identified [59]. 

Finally, the opening of the capsid in the nuclear basket could also be beneficial for the virus. Indeed, it has been shown that the nuclear basket serves as a platform for DNA repair, and is enriched for cellular factors involved in DNA repair [60,61,62]. Thus, the release of rcDNA in the nuclear basket could favor its repair, as well as help the virus to escape the cellular response that could be triggered if rcDNA was released directly in the nucleoplasm. Further studies are required to determine whether the nuclear pore basket is also important for cccDNA formation and for antiviral cellular response escape.

Alternatively, Lupberger and collaborators propose that the import of viral DNA is mediated by the viral polymerase, which contains a bipartite NLS [63]. Polymerase-mediated import would require the partial/complete exposure of the polymerase in the cytoplasm or at the cytoplasmic face of the NPC (Figure 2). The presence of an unstable or partially disassembled capsid in the cytoplasm remains a matter of debate. The group of J.T. Guo has observed the presence of DNase1-sensitive nucleocapsids (indicative of a more unstable/partially disassembled capsids) in the cytoplasm. The large majority of these capsids, according to their results, contain deproteinated rcDNA (dprcDNA), which they consider to be the precursor of cccDNA [42,59,60]. They suggest that the maturation processes inside the capsid (completion of the (+) strand DNA and the removal of the polymerase) in the cytoplasm lead to structural changes, allowing the exposure of the capsid NLS and nuclear import [46]. However, their data do not clearly establish whether the dprcDNA is the true cccDNA precursor or a dead-end product. Moreover, they have been challenged by others who observed that cytoplasmic capsids are mostly resistant to DNase digestion [11]. Further studies will be needed to determine where maturation steps leading to a functional cccDNA precursor take place. However, looking carefully at Guo and collaborators’ results, one cannot completely rule out the existence in the cytoplasm of destabilized/partially disassembled capsids containing polymerase bound rcDNA [46]. Indeed, in co-immunoprecipitation experiments using antibodies directed against either capsid C-terminal amino acids residues or karyopherin, dprcDNA recovered represents only a proportion of all of the precipitated DNase1 sensitive rcDNA, suggesting that part of this rcDNA is still bound to the viral polymerase. In this case, Pol NLS could also be involved in the nuclear import. Total disassembly of the capsid and the release of the viral DNA bound to the polymerase in the cytoplasm seems unlikely, as the viral DNA would be sensed by the innate immune system [64].

## 4. Conversion of rcDNA to cccDNA

rcDNA has a peculiar structure related to the way in which it is synthesized by the viral polymerase (Pol). First, in contrast to conventional reverse transcription, a Tyrosine residue (Tyr68) in the terminal protein (TP) domain of Pol acts as the acceptor for the first dNTP. This “protein priming” leads to the covalent linkage of Pol to the 5′ end of the (−) strand of rcDNA. The (+) strand DNA is primed by an 18-nucleotide-long capped RNA oligomer derived from the 5′ end of pgRNA, which remains associated with the 5′ end of the (+) strand in the mature rcDNA. The (+) strand DNA synthesis bridges the gap between the 5′ and 3′ end of the (−) strand DNA, generating a short triple-stranded region call “r”. Finally, the (+) strand DNA synthesis rests incomplete, leaving approximately half the genome single stranded prior to virion formation. Of note, all of the genome maturation steps after priming require the encapsidation of Pol and the viral genome into the capsid. In summary, rcDNA consists in a complete coding (−) strand covalently attached at its 5′ end to the viral polymerase and bearing a nine-nucleotide redundant sequence at its extremities, as well as a complementary incomplete (+) strand that is conjugated at its 5′ end to a capped RNA primer (Figure 3) [4,65]. cccDNA formation from rcDNA requires (i) the completion of the single stranded part of the viral genome, (ii) the removal of Pol from the 5′ of the (−) strand DNA, (iii) the removal of the capped RNA from the 5′ end of the (+) strand DNA, and, (iv) in addition, one strand of “r”—the redundant nine-nucleotide-long triple stranded region—has to be removed to allow for the ligation of the 5′ and 3′ ends of the two DNA strands (Figure 3). All of these steps require several enzymes, such as DNA polymerases, an endonuclease to remove the polymerase and the RNA primer, and a DNA ligase. 

Remaining unexplored for decades, mostly because of a lack of a manageable cellular model of infection, the mechanisms involved in cccDNA formation are starting to be better understood, but still require more investigation. Firstly, the mechanism involved in Pol removal remains debated. The group of M. Nassal first reported, using biochemical and silencing approaches, that the cellular tyrosyl-DNA phosphodiesterase 2 (TDP2) is the factor responsible for the removal of the viral polymerase [66]. However, different groups using silencing approaches were not able to reproduce these results [58,67]. Recently, a study aiming at characterizing the four termini of the cytoplasmic dprcDNA showed that the viral polymerase is removed through unlinking the tyrosyl-DNA phosphodiesterase bond between the polymerase and the (−) strand DNA by enzyme(s) that are yet to be uncovered [67]. However, as mentioned previously, it is not clear whether the cytoplasmic dprcDNA represents the true intermediate of nuclear cccDNA. 

Similarly, completion of the (+) strand has been a matter of debate. While studies have suggested that the viral polymerase is responsible for (+) strand completion, other studies using specific inhibitors have invalidated this hypothesis [68,69,70]. Instead, (+) strand HBV DNA completion is achieved through the DNA polymerase kappa (Pol κ) [70]. Recently, Wei and Ploss developed an in vitro system using recombinant rcDNA and cellular extracts from yeast or hepatoma cells to study cccDNA formation. They identified the cellular proteins involved in DNA lagging-strand synthesis and DNA repair—the proliferating cell nuclear antigen (PCNA), replicating factor C complex (RFC), and Pol delta (Pol δ) [71,72]. While the two studies identified two different polymerases, they are not necessarily contradictory, as studies have shown that both polymerases can work together during nucleotide excision repair (NER) [73]. Interestingly, Pol k is believed to be important for DNA repair when the pool of dNTPs is low, as in quiescent cells [74]. A recent study highlights the diversity of the mechanisms that could be involved in cccDNA synthesis, as Tang and colleagues have suggested that, contrary to cccDNA generated following de novo infection, the formation of cccDNA emanating from recycling required the activity of Pol α and Pol δ [75]. These intriguing results suggest that rcDNA may differ between the rcDNA coming from de novo infection and the rcDNA coming directly from recycling. Alternatively, for an unknown reason, capsids may use a different nuclear import pathway and thus the rcDNA may be released in different regions of the nucleus where different DNA repair proteins are enriched. The in vitro cccDNA formation assay also confirmed the involvement of a factor previously identified by others—the flap structure specific endonuclease 1 (FEN1). FEN1 has been shown to be involved in the cleavage of the redundant region “r” by using silencing approaches or a FEN1 inhibitor [76]. Interestingly, using an in vitro cccDNA formation assay, Kitamura and colleagues demonstrated that rcDNA can be efficiently converted into cccDNA using Bst DNA polymerase, Taq DNA ligase, and FEN1 [76]. Finally, this in vitro assay also confirmed the role of LIG 1. Indeed, using an shRNA screen covering 107 cellular repair genes, Long and collaborators previously identified the host cellular ligases 1 and 3 (LIG 1 and LIG 3) as the enzymes responsible for rcDNA termini ligation [69]. Their screening also identified two DNA repair proteins, APEX1 and PolB, that could be involved in cccDNA formation. In addition, using a screen based on the use of compounds that target cellular enzymes, Sheraz and colleagues identified topoisomerase 1 and topoisomerase 2 (TOP 1 and TOPO 2) as being involved in the circularization of viral DNA. However, the mechanism remains unknown and experiments using siRNA targeting TOP1 gave contradictory results [77]. While the exact kinetic of cccDNA formation is not yet clearly established, data from different laboratories suggest that the completion of the (−) strand is faster than the completion of the (+) strand, and occur independently [78,79].

One important question concerns the mechanism that marks rcDNA for the subsequent recruitment of the cellular enzymes responsible for cccDNA formation. Because of the damaged-DNA-like structure of rcDNA, Luo and colleagues assessed the role of two DNA damage repair pathways (DDR), ataxia telangiectasia mutated (ATM) and ataxia telangiectasia and RAD3-related (ATR), in cccDNA formation using inhibitors and silencing approaches. Their results supported the role of ATR in HBV cccDNA formation [80]. Because all of the results were generated mostly in hepatoma cells, one may argue that the expression of repair proteins and pathway activity in quiescent cells such as hepatocytes may differ from those in replicative cells, as was shown for neurons or myoblasts [81,82]. However, the liver may not completely fall in this category, as hepatocytes keep their ability to replicate. Moreover, the liver, because of its biological functions such as detoxification, is exposed to multiple DNA damage agents, which may require it to be especially proficient for DNA repair [83]. Nonetheless, studying cccDNA formation in transformed cells can have its own biases and limitations. Indeed, if transformation is linked to the disruption of a DDR pathway, in order to survive, the cells usually develop mechanisms to increase their repair capacity via another redundant repair pathway [84].

The cccDNA is organized as a mini-chromosome associated with histone and non-histone proteins, but the exact timing and factors involved in cccDNA chromatinization remains to be fully understood. Early studies have isolated HBV nucleoprotein complexes from the hepatoma cell line HepG2.2.15, with the four core histone proteins (H2A, H2B, H3, and H4), linker histone H1, and HBc associated with cccDNA [85,86]. In vitro experiments with *Xenopus laevis* oocyte extracts reconstituted the chromatin on HBV cccDNA and rcDNA, although only the cccDNA chromatin had a regular spacing and structure [86]. Interestingly, the authors found that the binding of HBc to the cccDNA is associated with a reduction in the spacing of the nucleosomes [86]. More recent studies with DHBV and HBV found that nucleosomes assembled into the cccDNA are not randomly positioned, and nucleosome-depleted regions were found in the Enhancer I and Enhancer II/BCP regions [87,88]. It remains unclear if chromatinization occurs after nuclear import in rcDNA, or if it takes place only after the repair and conversion of rcDNA to cccDNA.

The histones assembled into the cccDNA are post-translationally modified and regulate HBV transcription and replication, and the viral protein HBx is important for the establishment and maintenance of an active chromatin state in the cccDNA [89,90,91]. An analysis of the cccDNA chromatin revealed an enrichment for active histone modifications, particularly the acetylation of histone H3 and histone H4, and the trimethylation of histone H3 lysine 4 (H3K4me3), a lack of H3K27me3, and variable levels of the repressive mark H3K9me3 were found in samples from chronically infected HBV patients, suggesting a complex regulation taking place in chronic HBV infection [88,92]. More recently, the non-canonical histone variant H3.3 2 was identified as a positive regulator of HBV transcription [93]. For a detailed overview of the epigenetic and transcriptional regulation of HBV cccDNA, we suggest the following reviews [6,94].

Reverse transcription of the encapsidated pgRNA occasionally does not proceed properly, and leads to a double-stranded linear DNA (dslDNA) that can follow the same fate as the rcDNA-containing capsids (secretion or intracellular recycling) [4]. In the nucleus, the dslDNA HBV genome could be either inefficiently converted by nonhomologous end joining (NHEJ) into a non-functional cccDNA, or integrated into the cellular genome by NHEJ or microhomology-mediated end joining [69,95,96,97]. However, the integrated HBV DNA is replication-incompetent, but is believed to contribute to liver cancer development [98].

## 5. Conclusions

HBV represents a threat to global health and is responsible for more than 880,000 deaths each year. Current approved antiviral treatments for chronic hepatitis B include nucleo(t)ides analogues and interferon α control HBV replication, and improve liver function. However, they are not curative, as they do not eliminate the cccDNA, which is responsible for viral persistence. Little is still known about the size and the half-live of the cccDNA pool in HBV-infected hepatocytes. Most of our knowledge comes from the analysis of the cccDNA levels in animal or tissue culture models [99,100,101]. The level of HBV cccDNA analyzed in different in vitro and in vivo models varies, but ranges between 0.2 to 2 copies per cell [102]. In a recent established model of HBV infection allowing virus propagation, 5 to 12 copies of cccDNA were found per infected cell, with a half-life estimated to be approximately 40 days [10]. HBV cccDNA decay was also assessed in chronic hepatitis patients receiving nucleo(t)ide analogue-based therapies [103,104]. These studies show that cccDNA persists in chronic patients, despite therapy, and estimate its half-life to be at around 9.2 months. Part of the maintenance/replenishment of cccDNA pools can be attributed to persistent replication, which remains during treatment [103,105]. In a recent study, Huang and collaborators used the emergence of lamivudine resistant HBV mutants to study cccDNA turnover. They showed that cccDNA turnover is faster than previously suggested, with a half-life ranging from 5.6 to 21 weeks [106]. The authors suggest that the complete elimination of cccDNA could thus be reached with effective therapies that completely block the replenishment of the cccDNA pools. However, this study assesses the turnover of actively transcribing cccDNA, but cannot exclude the existence of a small pool of more or less silent cccDNA that can persist as a reservoir. While the mechanisms of cccDNA maintenance are not yet fully understood, the elucidation of all of the steps leading to cccDNA formation, from the entry of rcDNA to its repair and cccDNA expression, will allow for the identification of therapeutic targets in order to block the establishment of new molecules of cccDNA. Moreover, an understanding of the mechanisms that govern capsid translocation and rcDNA release, as well as the identification of cellular factors that interact with rcDNA during these early steps, may help to understand why HBV is not sensed by the innate immunity. The restoration of an antiviral response may help to eliminate HBV, in combination with additional antiviral treatments [107]. To date, several drugs are under development that directly target different stages of the HBV life cycle or modulate the innate or adaptative cellular antiviral response. One of these compounds, Bulevirtide, a synthetic N-acetylated peptide derived from HBsAg, blocks HBV entry and has already been approved in the European Union for the treatment of chronic hepatitis delta virus infection. Compounds that target nucleocapsid assembly, called capsid assembly modulators (CAMs), represent an area of extensive research, and CAMs are already in preclinical trial [108]. However, the consensus in the field is that drugs will have to be used in combination in order to achieve an HBV cure. New therapeutic approaches and the results of preclinical/clinicals trials have been discussed in recent reviews [107,109,110].

## Figures and Tables

**Figure 1 viruses-13-00757-f001:**
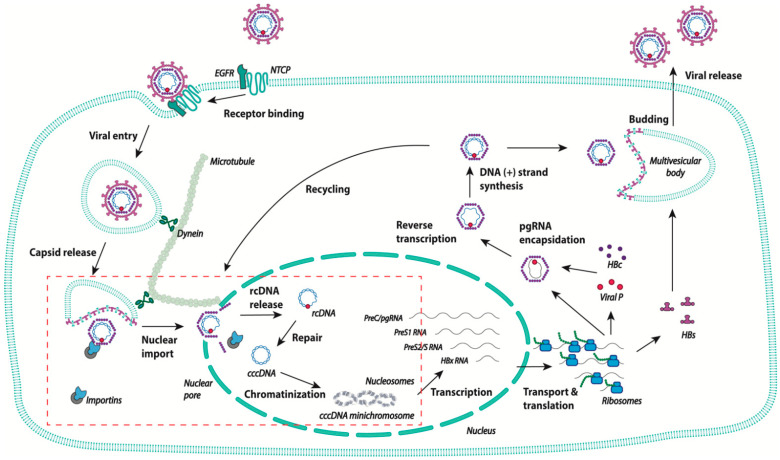
Schematic representation of the human hepatitis B virus (HBV) life cycle. After binding to the high affinity receptor sodium taurocholate co-transporting polypeptide (NTCP), HBV is internalized via endocytosis, and the viral capsid is released into the cytoplasm. The capsid travels to the nuclear pore using the microtubule network. The rcDNA is released in the nucleus and is converted into covalently closed circular DNA (cccDNA), which becomes chromatinized, and is the template for the transcription of all HBV RNAs. In the cytoplasm, pgRNA is encapsidated and retrotranscribed into rcDNA. The capsids are then either enveloped at MVB and secreted as new virions, or transported back to the nuclear pore to increase the pool of cccDNA. The steps of the viral life cycle discussed in the review are delineated in the red box.

**Figure 2 viruses-13-00757-f002:**
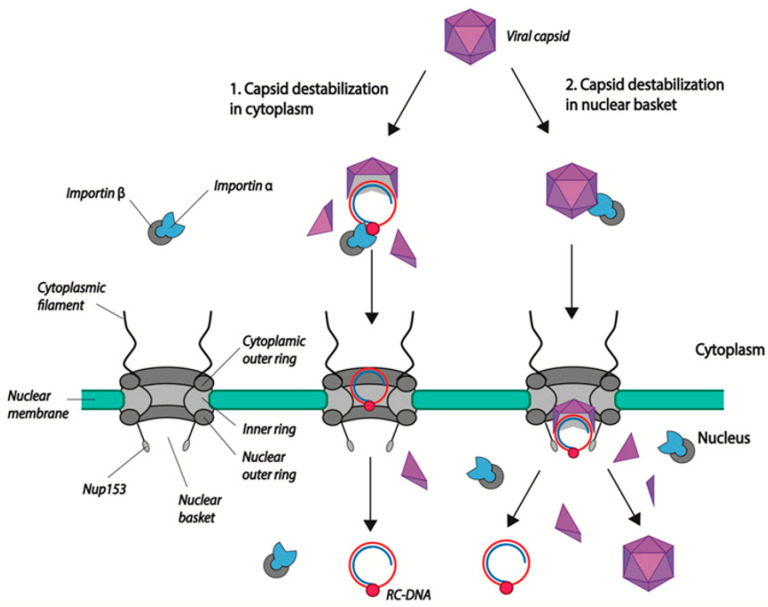
Models of HBV genome nuclear import. (1) Capsid disassembly/partial disassembly occurs in the cytoplasm (or at the nuclear pore) leading to the exposure of the nuclear localization signal (NLS) in the terminal protein (TP)-domain of Pol, and the subsequent binding to nuclear transport factors (importin α and β). (2) Capsids are transported into the nuclear basket following the interaction of NLS present in the core CTD with the import factors importin α/β. There, capsids bind directly to Nup153 and mature capsids disintegrate, leading to the release of the viral genome. Capsid dimers enter the nucleus where they can re-assemble. rcDNA is represented as attached to the viral polymerase (red circle). Capsid dimers are represented as purple triangles.

**Figure 3 viruses-13-00757-f003:**
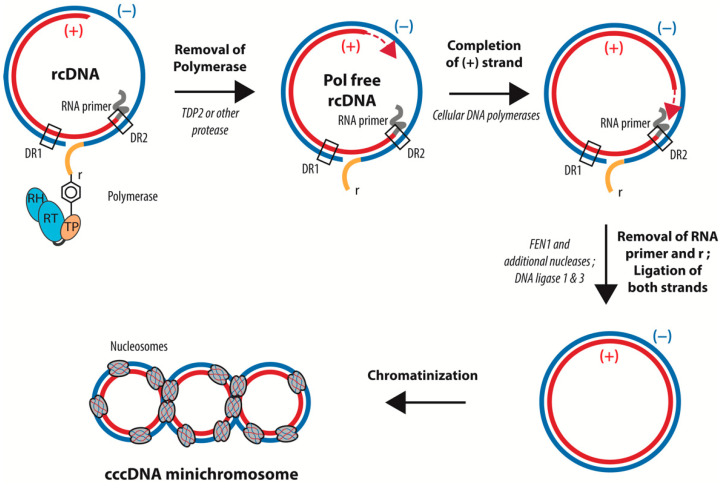
Schematic representation of the conversion of rcDNA into cccDNA. cccDNA formation requires: (i) the removal of Pol from the 5′ of the (−) strand DNA, (ii) the completion of the (+) strand, (iii) the removal of the capped RNA from the 5′ end of the (+) strand DNA, (iv) the cleavage of the redundant sequence “r”, and (v) the ligation of the 5′ and 3′ ends of the two DNA strands. cccDNA is then loaded with histones. The precise time sequence of these different steps is not yet completely determined.

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
