# Peer review of "Early Steps of Hepatitis B Life Cycle: From Capsid Nuclear Import to cccDNA Formation"

_viruses, 2021, doi:10.3390/v13050757_

Round 1
Reviewer 1 Report
In this paper, Dias et al. review current knowledge on early steps of HBV life cycle, focusing on nuclear import and conversion of the rcDNA into cccDNA. There are still many grey areas for these steps addressed by the authors. Here are some suggestions to improve the manuscript.
«Replication» in the title should be replaced by «life cycle». Keep the word replication for the ad-hoc step that is blocked by nucs.
I would recommend adding a paragraph reviewing the chromatinization steps of the cccDNA. If authors chose not to, it should be stated in the abstract and the introduction that the paper will not review this step.
English needs extensive proofreading and correction by a native speaker.
Exemples but not exhaustive:
line 89 «to the formation of the cccDNA» («to cccDNA formation»)
line 150 capsid(s), 177 dimer(s), 276 dNTP(s), 319 RNA(s)
line 201 have been/has been, 202 Giving/given... 205 required/requires, 208 enriched for, 209 help to escape, 285 involvement of a factor
Please, homogenize polymerase description between figures and text (p / pol / POL...)
line 32 add mention that this takes place within the capsids
line 43 to 57 from "rcDNA" to "figure 2" and figure 2 could probably fit better at the beginning of section 4 for more clarity.
line 58 please change "covalent closed DNA" by "covalently closed circular DNA"
line 63 please clarify if mechanisms regulating the fate of nucleocapsids between secretion and recycling are known and if they happen differently in each phase of the disease.
Figure 1: highlight with a box the steps rewieved in the current paper
add recycling of capsid-rcDNA to the nucleus (as mentioned in the legend)
budding is not adequately mentioned in the figure
Add question marks to the debated steps
In the present form Figure 1 doesn’t bring much to the field. It would be of better interest to detail more the steps described in the review (ex mention of the EGFR, clathrin, importins...) and less the other ones (transcription/translation/release).
Paragraph 3
line 140 mention of the coordinates of the NLS would be appreciated
lines 142-143 Are there some mutations impacting cellular localization found in patients and/or specific genotypes?
line 156 which type of RNA?
lines 152-173 illustrations of the 3D structures of HBc with phosphorylated and unphosphorylated CDT could be helpful. If structural information still lacking please state it.
lines 158-166 is there a competition for docking between mature HBc particles and empty/RNA containing capsids?
Do certain physiological conditions such as fasting affect the HBc phosphorylation state? Please elaborate.
line 205 that remain to be identified
line 218 needs some editing: gnome
Before line 139 precise that rcDNA import mechanisms remains controversial with several model proposed. Capsid NPC docking / import via viral polymerase. For more clarity and coherence with Figure 3, I would suggest placing paragraph 220-244 first.
Paragraph 4 Please discuss double stranded linear DNA and if HBx RNA found in viral particules (Niu et al 2017) has a role in the establishment of cccDNA.
Figure 2 it would be helpful for the reader to detail more the different steps mentioned in this paragraph with mention of the proteins involved at each step. Chromatinization and minichromosome can be removed from this figure as they are not reviewed here. They can make a specific figure if the authors decide to detail these steps.
line 275 needs some editing: pol k typo
line 306 needs some editing: missing word after cccDNA
lines 318-319 histones loading is part of the establishment of the cccDNA. A short review of what is currently known would be appreciated.
Conclusion is pretty vague and should be more consistent. Mentions of the timing of cccDNA establishment after infection, range of cccDNA molecules per infected hepatocyte would be appreciated. When you say «While the mechanisms of cccDNA maintenance are poorly 325 understood, elucidation of all the steps leading to cccDNA formation from the entry of rcDNA to its repair and cccDNA expression will allow the identification of therapeutic targets in order to block the establishment of new molecules of cccDNA». As the review is precisely about these steps, can you sum up which therapeutic targets you think are of interest? Likewise, could you elaborate on the sentense 329-30 and make hypothesis from what you described in paragraph 3.
Author Response
1- English needs extensive proofreading and correction by a native speaker.
The text has now been reviewed by a native English speaker. Moreover, typos pointed out by the reviewer have been fixed.
2-The Please, homogenize polymerase description between figures and text (p / pol /POL).
We homogenized using Pol.
3- line 32 add mention that this takes place within the capsids
We modified the text and wrote line 32 : “of the encapsidated pregenomic RNA (pgRNA)”
4- line 43 to 57 from "rcDNA" to "figure 2" and figure 2 could probably fit better at the beginning of section 4 for more clarity.
We acknowledge the reviewer point. We moved the figure 2 (now figure 3 in the revised version) at the beginning of section 4.
5- line 58 please change "covalent closed DNA" by "covalently closed circular DNA"
We made the correction.
6- line 63 please clarify if mechanisms regulating the fate of nucleocapsids between secretion and recycling are known and if they happen differently in each phase of the disease.
To our knowledge the mechanism leading to nucleocapsids recycling is not fully understood. For Duck HBV it has been linked to the level of envelop proteins, but this is less clear for HBV. We modified the sentence on line 52 to clarify this point.
Indeed, depending of the inflammation state of the infected patients, most capsids are located either in the cytoplasm or in the nucleus but to my knowledge is not correlated with the amplification of the pool of cccDNA.
7- Figure 1: highlight with a box the steps rewieved in the current paper, add recycling of capsid-rcDNA to the nucleus (as mentioned in the legend), budding is not adequately mentioned in the figure, Add question marks to the debated steps. In the present form Figure 1 doesn’t bring much to the field. It would be of better interest to detail more the steps described in the review (ex mention of the EGFR, clathrin, importins...) and less the other ones (transcription/translation/release).
We modified Figure 1 accordingly. For clarity purpose, however we did not add question marks to debated steps (questions remain for almost all the steps). We highlighted with a red box the steps reviewed in the current paper.
Paragraph 3
8- line 140 mention of the coordinates of the NLS would be appreciated
We acknowledge the reviewer comment and added a sentence on line 142 to clarify this point. We could not give the exact coordinates of the putative NLS because they are still debated but we did mention the localization and cited the 3 papers that describe the NLSs.
9- lines 142-143 Are there some mutations impacting cellular localization found in patients and/or specific genotypes?
The CTD is highly conserved between all the genotypes. In their review published in Viruses vol 12 (2020), Hugues de Rocquigny and collaborators compared in Figure 5 the CTD sequences of different genotypes and observed a strong conservation of the CTD. Beside, analysis of mutations in Precore/core regions in chronically infected patients showed that mutations in the CTD are very sparse, mutations occuring mainly in the immunoreactive regions of the core protein (Tcell and B cell epitopes). These mutations likely emerge to alter core antigenicity (Kim et al;, Plos One 2012, Zare-BiDaki et al., Clin. Lab. 2014; Al-Qahtani et al., Frontiers in Cellular and infection Microbiology 2018).
10- line 156 which type of RNA?
We modified the text and specified : pgRNA line 148.
11- lines 152-173 illustrations of the 3D structures of HBc with phosphorylated and unphosphorylated CDT could be helpful. If structural information still lacking please state
We agree with the reviewer that illustration of the 3D structures of HBc with phosphorylated and unphosphorylated CDT could help the reader. So, we now specifically refer in the text line 148 and line 166 to the publications of the team of Adam Zlotnick that describe electron cryo-micrographs analysis of empty or pgRNA-filled particles using unphosphorylated core proteins or phosphorylation-mimic core protein.
12- lines 158-166 is there a competition for docking between mature HBc particles and empty/RNA containing capsids?
According to published data, CTDs in pgRNA filled capsids are located towards the interior of the capsid. Thus, theoretically, these capsids should not compete with the mature capsids. In accordance, Rabe and collaborators (2003) showed using immunofluorescence that capsids derived from cells treated with an inhibitor of reverse transcription (containing thus pgRNA) do not localize at nuclear pores. However, CTDs in empty capsids are exposed to the exterior of the capsids as in mature capsids. Chen and collaborators showed however that empty capsids bind preferentially importin b, suggesting that these capsids use a different trafficking pathway for nuclear import (Chen et al., 2016).
To clarify these points, we modified the text lines 166-174 section 3.
13- Do certain physiological conditions such as fasting affect the HBc phosphorylation state? Please elaborate?
Liver metabolism plays without question an important role in HBV replication. HBV infection has been associated for example with lipid metabolism changes which in turn may favor viral replication (review in Zhang et al., 2021). The impact of fasting on HBV replication has been studied by different groups and results show that fasting increases virus transcription in part through up-regulation of PGC1-a (Shlomai et al., 2006,Li et al., 2009). We are not aware however of studies analyzing the impact of fasting on Hbc phosphorylation.
14- line 205 that remain to be identified
We acknowledge the reviewer comment and modified the sentence line 205.
15- line 218 needs some editing: gnome
Fixed.
16- Before line 139 precise that rcDNA import mechanisms remains controversial with several model proposed. Capsid NPC docking / import via viral polymerase. For more clarity and coherence with Figure 3, I would suggest placing paragraph 220-244 first.
We acknowledge the reviewer comment and added a sentence in the text line 127 specifying that the rcDNA import mechanisms remain controversial and that two models are proposed.
Paragraph 4
17- Paragraph 4 Please discuss double stranded linear DNA and if HBx RNA found in viral particules (Niu et al 2017) has a role in the establishment of cccDNA.
We acknowledge the reviewer comment and discussed the fate of double stranded DNA-filled capsids in the text section 4 lines 365-372.
We decided here not to discuss the role of RNA-containing capsids because the topic of this review was the nuclear import of the viral genome. Besides, capsids containing HBx RNA, if involved in HBV life cycle, may not impact cccDNA establishment but rather allow HBV cccDNA transcription.
18- Figure 2 it would be helpful for the reader to detail more the different steps mentioned in this paragraph with mention of the proteins involved at each step. Chromatinization and minichromosome can be removed from this figure as they are not reviewed here. They can make a specific figure if the authors decide to detail these steps
We acknowledge the reviewer comment and detail the chromatinization of the cccDNA in section 4 lines 340 to 372.
Enzymes are indicated on figure 3 (former figure 2).
19- line 275 needs some editing: pol k typo
Fixed.
20- line 306 needs some editing: missing word after cccDNA
Fixed.
21- Conclusion is pretty vague and should be more consistent. Mentions of the timing of cccDNA establishment after infection, range of cccDNA molecules per infected hepatocyte would be appreciated. When you say «While the mechanisms of cccDNA maintenance are poorly 325 understood, elucidation of all the steps leading to cccDNA formation from the entry of rcDNA to its repair and cccDNA expression will allow the identification of therapeutic targets in order to block the establishment of new molecules of cccDNA». As the review is precisely about these steps, can you sum up which therapeutic targets you think are of interest? Likewise, could you elaborate on the sentense 329-30 and make hypothesis from what you described in paragraph 3
We acknowledge the reviewer comment that rejoins the comment of reviewer 2. We modified the conclusion and provided a quick overview of cccDNA quantification and turnover in infected hepatocytes. We also discussed briefly current therapeutic developments with regard to targets of interest.
Reviewer 2 Report
I red with great interest the manuscript by Dias and colleagues describing in detail the molecular events from HBV entry into the hepatocyte to the formation of the supercoiled HBV DNA. Overall, the manuscript is well written and provides a careful virologic revision of the mechanism underlying HBV infection; an accurate knowledge of such mechanism is absolutely important for the development of novel drugs able to directly target (degrading/silencing) the HBV cccDNA, which is responsible of HBV persistence within infected hepatocytes. Indeed, as the authors mentioned in the manuscript, currently we are able to control viral replication and thus the liver disease, but the definite elimination of the virus from the liver is far from being achieved.
Below, some minor comments:
1) Figure 1. To provide an almost complete cycle of HBV, authors should add in the picture the recycling of rcDNA into cccDNA. Indeed, in the legend of figure 1, it is stated that "the capsids are then either enveloped at MVB and secreted as new virions or trasnported back to the nuclear pore to increase the pool of cccDNA".
2) Line 70. The authors correcly state that "different types of incomplete particles are released particularly empty capsids...". Given the highly virologic and modelcular fingerprint of the manuiscript, it should be mentioned the caracteristics of the HBcAg that lead to the formation of empty capsids. This aspect is important also considering that "novel" biomarkers (i.e. HBcrAg) that are entering into clinical practice rely on this ferature.
3) Authors might consider to spend some words on novel therapies under investigation and the corresponding molecular targets. In my opinion, at least a paragraph (5-6 lines) on the inhibitor of HBV entry should be added in the manuscript (considering also its recent approval by the EMA for the treatment of HDV).
Author Response
1) Figure 1. To provide an almost complete cycle of HBV, authors should add in the picture the recycling of rcDNA into cccDNA. Indeed, in the legend of figure 1, it is stated that "the capsids are then either enveloped at MVB and secreted as new virions or trasnported back to the nuclear pore to increase the pool of cccDNA".
Fixed.
2) Line 70. The authors correcly state that "different types of incomplete particles are released particularly empty capsids...". Given the highly virologic and modelcular fingerprint of the manuiscript, it should be mentioned the caracteristics of the HBcAg that lead to the formation of empty capsids. This aspect is important also considering that "novel" biomarkers (i.e. HBcrAg) that are entering into clinical practice rely on this ferature
To acknowledge the reviewer comment, we modified the section 1 lines 63-66. We emphasized the importance of these particles as potential biomarkers and cited publications that discuss their characteristics as well as their used as biomarkers.
3) Authors might consider to spend some words on novel therapies under investigation and the corresponding molecular targets. In my opinion, at least a paragraph (5-6 lines) on the inhibitor of HBV entry should be added in the manuscript (considering also its recent approval by the EMA for the treatment of HDV).
We acknowledge the reviewer comment that rejoins the comment of reviewer 1. We briefly commented on novel therapies in development with regard to potential targets and discussed more specifically on Bulevirtide in section 5 lines 403-412.